# Stress-Induced Changes in the Endogenous Opioid System Cause Dysfunction of Pain and Emotion Regulation

**DOI:** 10.3390/ijms241411713

**Published:** 2023-07-20

**Authors:** Kazuo Nakamoto, Shogo Tokuyama

**Affiliations:** Department of Clinical Pharmacy, School of Pharmaceutical Sciences, Kobe Gakuin University, 1-1-3 Minatojima, Chuo-ku, Kobe 650-8586, Japan

**Keywords:** opioid, stress, pain, emotion, fatty acid receptor

## Abstract

Early life stress, such as child abuse and neglect, and psychosocial stress in adulthood are risk factors for psychiatric disorders, including depression and anxiety. Furthermore, exposure to these stresses affects the sensitivity to pain stimuli and is associated with the development of chronic pain. However, the mechanisms underlying the pathogenesis of stress-induced depression, anxiety, and pain control remain unclear. Endogenous opioid signaling is reportedly associated with analgesia, reward, addiction, and the regulation of stress responses and anxiety. Stress alters the expression of various opioid receptors in the central nervous system and sensitivity to opioid receptor agonists and antagonists. μ-opioid receptor-deficient mice exhibit attachment disorders and autism-like behavioral expression patterns, while those with δ-opioid receptor deficiency exhibit anxiety-like behavior. In contrast, deficiency and antagonists of the κ-opioid receptor suppress the stress response. These findings strongly suggest that the expression and dysfunction of the endogenous opioid signaling pathways are involved in the pathogenesis of stress-induced psychiatric disorders and chronic pain. In this review, we summarize the latest basic and clinical research studies on the effects of endogenous opioid signaling on early-life stress, psychosocial stress-induced psychiatric disorders, and chronic pain.

## 1. Introduction

Unfavorable nurturing experiences in childhood, such as neglect and physical, sexual, and psychological abuse, cause early life stress and are considered serious social problems. They can cause psychiatric disorders, such as depression, anxiety, and personality disorders in adulthood [1,2]. In addition, social life events, such as interpersonal difficulties and conflicts, defeat due to unemployment and heartbreak, and the death of relatives, can be psychological and social stressors. Catastrophic events that strongly impact life, such as natural disasters, war, and crime, also cause post-traumatic stress disorder [3]. Thus, exposure to chronic psychological and social stress may be a risk factor for the development of psychiatric disorders.

Furthermore, repetitive stress exposure has been reported as a risk factor for the dysfunction of pain regulatory mechanisms, including chronic pain [4,5,6,7]. For example, preoperative anxiety leads to a state of catastrophic thinking that negatively perceives the experience of pain and is a clinically significant problem for many patients undergoing surgery. High anxiety levels may be associated with prolonged and exacerbated post-operative pain [8]. Currently, there is no treatment for stress-induced pain exacerbation and/or chronicity because the existing analgesic drugs act symptomatically to relieve pain. Therefore, there is an urgent need to identify target factors that are strongly affected by psychological and social stress and conduct drug discovery research to improve the therapeutic efficacy of drugs for stress-induced chronic pain, which might also provide valuable information for understanding the pathological mechanisms of stress-induced chronic pain.

The endogenous opioid system plays an important role in pain [9]. This system consists of four seven-transmembrane G-protein-coupled receptors (GPCRs): Mu, delta, kappa, and nociceptin (MOR, DOR, KOR, and NOR) [10,11]. Each receptor is encoded by a unique gene (*OPRM1*, *OPRD1*, *OPRK1*, or *OPRL1*). These receptors share more than 60% of their amino acid composition. Several opioid receptor agonists, such as morphine, hydromorphone, tapentadol, oxycodone, and fentanyl, have been developed and are being used clinically worldwide as medical narcotics because of their potent analgesic effects. Recently, the crystal structures of these receptors were elucidated, leading to accelerated drug discovery research targeting opioid receptors [12,13,14,15]. These receptors are activated by endogenous opioid peptides, such as endorphins, enkephalins, dynorphins, and nociceptin/orphanin FQ (N/OFQ), under physiological conditions [16,17]. Moreover, opioid signaling is closely related to emotions and is strongly affected by stress exposure [9,11]. Existing evidence indicates that endogenous opioid signaling is highly affected by exposure to adverse childhood experiences, such as child abuse, neglect, and psychological and social stress (interpersonal relationship problems, unemployment, social defeat and frustration, and loss of immediate family members) [18,19,20]. Thus, dysfunction of the endogenous opioid system may be a risk factor for psychiatric disorders and chronic pain.

In this review, we summarize the changes in the endogenous opioid system and the dysfunction of emotion and pain regulatory mechanisms induced by various stress exposures, such as early life and psychosocial or social stress.

## 2. Early Life Stress

Recently, “maltreatment” has been proposed as a wide-ranging concept encompassing neglect and/or experiences of any emotional, physical, or sexual abuse, possibly harming and affecting the individual’s physical and mental health. In addition, a stressful environment in early life may lead to various mental illnesses in childhood, such as depression, anxiety, psychosis, and personality disorders [21,22]. However, the molecular and pathological mechanisms underlying the development of stress vulnerability at maturity after early life stress (ELS) exposure remain unclear, and there are no effective treatments for ELS-induced psychiatric disorders.

Several epidemiological studies have shown that exposure to ELS is associated with the development of psychiatric disorders in adulthood. For example, Romanian adoptees exposed to ELS for more than six months showed higher rates of autism spectrum disorder, disinhibited social engagement, and inattention and overactivity through young adulthood than controls from the UK [23]. Similarly, in the largest cross-sectional study of adults in the US, 62% of respondents had experienced at least one adverse childhood experience (ACE), and approximately one-quarter had experienced three or more ACEs [24]. These results indicate that the greater the number of ACEs, the greater the risk of developing mental illness [25]. Over the past few years, other factors, such as alcohol abuse and drug problems, have been identified as risk factors for ELS. In particular, opioid abuse and misuse may be responsible for a large proportion of the ELS cases, as the opioid crisis is a substantial social problem in the US [26]. Therefore, there is an urgent need to formulate treatment and prevention strategies for ACE.

### 2.1. Influence of the Opioid Signal on ELS

MOR signaling is involved in pain control and plays an important role in regulating social emotions and behavior [27,28]. MOR-deficient mice exhibit behavioral phenotypes, such as reduced maternal attachment, decreased social interaction, and exacerbation of anxiety, which are the core and secondary symptoms of autism spectrum disorder [29,30,31]. In addition, the child MOR variant (*OPRM1* A118G) was associated with improved parent–child relations, suggesting influences on parent–child relationships [32]. Thus, opioid signaling is important in the development of the parent–child relationship during childhood. However, this signaling might be strongly affected by ELS, and the alteration of its function may be a key step in inducing mental illness. Therefore, ELS exposure may alter opioid receptor expression and downstream signaling, affecting the sensitivity, affinity, and binding capacity of opioid agonists. Socially isolated rats during adolescence exhibited greater ethanol intake and preference through an increase in KOR function in the nucleus accumbens (NAc) [33,34]. In postmortem samples from depressed individuals who died by suicide, with or without a history of severe child abuse, and from psychiatrically healthy control subjects, a history of child abuse was explicitly associated with the downregulation of KOR in the anterior insula [18]. Thus, KOR may be epigenetically regulated by stressful childhood experiences. Chang et al., demonstrated a sex difference in KOR expression in the nucleus accumbens of rats subjected to neonatal predator odor exposure at neonatal and juvenile time points [35]. Michaels and Holtzman showed that rats with ELS induced by maternal separation showed a higher place preference in the conditioned place preference test for morphine, a MOR agonist, than controls, but not for the KOR agonist, and showed a lower place aversion to spiradoline, a selective κ-opioid agonist. These results raise concerns that ELS may increase opioid abuse by increasing sensitivity to the rewarding effects of opioids while decreasing KOR-mediated aversive effects [34]. Vazquez et al., reported that rats subjected to maternal deprivation stress for 3 h per day for 14 days after birth exhibited hypersensitivity to the reinforcing effects of morphine, development of morphine and sucrose dependence, and hypoactivity of the enkephalinergic system as adults [36]. As far as we know, there are few reports regarding the relationship between ELS and NOR. The Western diet and environmental stress exposure to the mother during gestation are considered risk factors for cognitive dysfunction in adulthood. N/OFQ in the hippocampus is important in the regulation of cognitive function [37]. NOR agonists impair learning and memory [38], whereas the inhibition of NOR activation has been associated with memory enhancement [39]. Therefore, the increased hippocampal *Oprl1* mRNA and *Oprl1* variants in the exposure to stress during gestation are involved in cognitive dysfunction in adulthood [40].

These results suggest that ELS alters the expression pattern of each opioid receptor in the brain and changes morphine responsiveness.

Previously, we reported that ELS model mice induced by maternal separation and social isolation (MSSI) showed emotional disorders such as depression-like behavior and anxiety-like behaviors, and chronic pain [41,42]. Pups in both groups (MSSI and control) were housed with dams until postnatal day 14. On postnatal day 15, the pups in the MSSI group were placed in individual isolation cages for 6 h per day away from their mothers. After a seven-day separation from their mothers, the pups were kept in isolation cages until they were 10 or 18 weeks old. These mice showed abnormal behavioral phenotypes, such as anxiety- and depression-like behaviors and altered social interactions [41]. We determined that MOR, DOR, and KOR mRNA expression decreased in the periaqueductal gray matter (PAG) of MSSI mice. In contrast, the expression of KOR mRNA was significantly increased in the amygdala of MSSI mice [19]. MSSI mice also demonstrated decreased morphine responsiveness over two months after the last ELS loading. Morphine modulates spinal nociceptive transmission by acting on opioid receptors in the supraspinal region and activating the endogenous descending pain inhibitory systems. The PAG is the primary site of action of opioid analgesics, such as morphine [43,44]. These results suggest that opioids in the central nervous system are strongly affected by ELS exposure, which attenuates the antinociceptive effects of morphine (Figure 1). Other studies have shown that the antinociceptive effects of morphine are reduced in mice subjected to other types of chronic stress, such as repetitive restraint stress or unpredictable chronic stress [45,46,47,48]. Therefore, ELS may significantly alter the responsiveness of opioid receptor agonists, such as morphine, resulting in altered pain sensitivity. Furthermore, the functional disruption of opioid signaling caused by ELS exposure may contribute to the development of psychiatric disorders and chronic pain.

### 2.2. Influence of ELS on Alteration of Pain Sensitivity and Chronic Pain

There is growing evidence that traumatic experiences during childhood influence pain sensitivity and the development of chronic pain in adulthood [49]. For example, child abuse increases the risk of developing fibromyalgia in adulthood [50,51]. In addition, children hospitalized following a road traffic accident and those who experienced maternal death and familial financial hardship have an increased risk of developing widespread chronic pain in adulthood [5]. In a recent centers for disease control and prevention (CDC) report, most patients with chronic pain experienced at least one ACE, demonstrating a significant difference between the analyzed groups. Higher maternal ACE scores corresponded with depressive symptoms and greater anxiety, greater sleep disturbances and fatigue, and greater pain intensity in mothers. [52], and the incidence of chronic pain increased two-fold in patients with ACEs (8.7%) compared with that in patients without ACEs (4.6%) in another study [2]. In a meta-analysis, children who experienced childhood abuse and neglect had worse pain symptoms than those who did not experience ELS [53]. Furthermore, those exposed to childhood stress have increased pain sensitivity and severity and pain incidence [53]. Therefore, excessive ELS exposure during childhood may increase the risk of developing chronic pain.

We have previously reported that MSSI mice with partial sciatic nerve ligation experienced an exacerbation of neuropathic pain compared to non-stressed mice [41]. Burke et al., claimed that 24-h mother-infant separation on postnatal day 9 exacerbated neuropathic pain in female mice during maturation [54]. ELS is known to exacerbate not only neuropathic pain but also fibromyalgia [55] and abdominal pain, which are associated with irritable bowel syndrome (IBS) [56,57]. However, there are few reports on the involvement of the altered function of opioid receptors and changes in their receptor expression in the mechanism for ELS-induced fibromyalgia and IBS-induced abdominal pain. Based on these and other reports, it can be concluded that early-life stressful events exacerbate neuropathic pain in adulthood in a rodent model.

## 3. Psychosocial Stress and Psychiatric Disorders

Health, safety, and well-being are protected in modern society through sophisticated medical care that promotes longevity. In addition, advances in science and technology have led to the development of an information-driven society, making life more convenient and comfortable. However, the increasingly competitive and controlled society and recent behavioral restrictions caused by the coronavirus disease pandemic have exposed people to psychological and social stress. Stress exposure adversely affects health, causing psychiatric disorders and chronic pain. Recently, the repeated social defeat stress model, which substantially reflects the symptoms of affective disorders induced by stress exposure, has been widely used in depression and anxiety research [58,59]. This section summarizes previous reports on the relationship between the social defeat stress mouse model and opioid signaling.

In the social defeat stress model, aggressive and large Institute of Cancer Research (ICR) mice were used as stressors. Experimental C57BL6J mice cohabitated in the home cage of ICR mice and were repeatedly exposed to direct aggression and indirect threats by the ICR mice. These mice exhibited anxiety- and depression-like behaviors and were averse to social contact with mice of the same species as the stressor. This mouse model is considered a good tool for studying how repeated exposure to social stresses, such as bullying or physical abuse, in humans increases the risk of depression [58]. This behavioral abnormality was not improved by a single administration of the tricyclic antidepressant imipramine or the serotonin reuptake inhibitor (SSRI) fluoxetine. However, it was reversed by their administration for four weeks after chronic defeat stress. Therefore, it is considered a useful model for elucidating the pathogenetic mechanisms of depression [59].

### 3.1. Influence of Social Defeat Stress on Opioid Signaling

#### 3.1.1. Social Defeat Stress and MOR

The endogenous opioid system is involved in reward and analgesia and is strongly influenced by stress. For example, MOR-deficient mice did not exhibit socially aversive behavior after social defeat stress [60], and mice with the *Oprm1* A112G SNP, the gene encoding MOR, showed stress vulnerability to social defeat stress. These findings indicate that MOR is important for stress-induced long-term neuroplastic changes [61]. Nikulina et al., reported increased MOR expression in the lateral ventral tegmental area (VTA) within 30 min of social defeat stress, which persisted for 21 days after social defeat stress, indicating that repeated social stress alters VTA dopaminergic function by increasing MOR expression [62,63]. Similarly, Jonston et al., showed that lentivirus-mediated knockdown of VTA *MOR* ameliorated the reduced social behavior and depression-like behavior associated with social defeat stress [64]. However, it is not clear which cells express MOR in the VTA region. Kudo et al., reported that 74.1% of MOR mRNA-expressing neurons in the VTA were gamma-aminobutyric acid (GABA) ergic, with the rest being glutamatergic neurons expressing type-2 vesicular glutamate transporter mRNA [65]. VTA MORs are presynaptically expressed by GABA neurons [66,67] and, when activated, reduce GABAergic inhibition of VTA DA neurons [68] in the VTA. Social stressors may function to increase endogenous MOR activity on GABA neurons, thus reducing the GABAergic inhibition of local DA neurons and facilitating behavioral sensitization to psychostimulant drugs. Further study will be needed to clarify which cells express MOR in the VTA using genetic MOR-Cre mice.

Psychological stressors stimulate the hypothalamic–pituitary–adrenal axis and sympathetic nervous system, releasing various hormones and neurotransmitters. However, the mechanisms underlying these phenomena are poorly understood. Previously, human and primate studies reported that MOR agonists suppress the hypothalamic–pituitary–adrenocortical (HPA) axis [69,70], whereas rodent studies showed that MOR agonists increased plasma adrenocorticotropic hormone and corticosterone levels [71]. There seem to be numerous contradictions in the literature regarding the effects of opioids on the HPA axis. MOR conditional knock-out mice subjected to chronic restraint stress demonstrated reduced activity of the HPA axis associated with the stress response and improvement in symptoms such as stress-induced depression-like behavior [72]. Thus, MOR may play an important role in reward-aversion processing and promote sociability and resilience to stress.

#### 3.1.2. Social Defeat Stress and DOR

DOR is involved in emotions, such as anxiety and depression, and learning and memory, as it is highly expressed in the limbic region of the central nervous system [73]. Previously, DOR agonists showed anxiolytic effects, whereas DOR antagonists or DOR-deficient mice exhibited enhanced anxiety-like behaviors [74]. DOR agonist SNC80 or KNT-127 decreased immobility time in the forced swim test [75,76], indicating that DOR plays an important role in emotions such as depression, anxiety, and fear. The endogenous opioid peptide leucine-enkephalin binds to two opioid receptors: MOR and DOR. Enkephalin is involved in antinociception and stress-induced analgesia and plays a major role in stress processes independent of its analgesic functions. Enkephalin-deficient mice exhibit enhanced anxious behavior. Berube et al., and Henry et al., suggested that enkephalins are important in regulating emotions and affective behavior and may be important regulators of stress resistance [20,77]. Enkephalin mRNA expression is decreased in the basolateral amygdala (BLA), and also DOR mRNA is decreased in the ventral hippocampal CA1 region of stress-vulnerable mice after social defeat stress exposure. It is thought that decreased expression levels of enkephalin and DOR mRNA are associated with vulnerability to social stress. Thus, activating the enkephalin/DOR signaling pathway may have a preventive effect on stress vulnerability caused by psychological stress [78]. Nam et al., reported that mice susceptible to social defeat stress showed reduced enkephalin levels and increased enkephalinases in the NAc, suggesting that enhancing enkephalins or DOR signaling in the NAc may effectively prevent reduced motivational behavior after social defeat stress [79]. Yoshioka et al., showed that repeated chronic vicarious social defeat stressed mice with a selective DOP agonist, KNT-127, improved their decreased social interaction behaviors and increased serum corticosterone levels [80].

#### 3.1.3. Social Defeat Stress and KOR

KOR activation produces both aversive and psychotomimetic effects, and its endogenous opioid peptide, dynorphin, plays an important role in stress regulation [81,82]. McLaughlin et al., showed that repeated social defeat stress for three days caused stress-induced immobility and analgesia. In contrast, KOR antagonist nor-binaltorphimine (nor-BNI)-injected mice and prodynorphin gene-disrupted mice did not show these behaviors [83]. Wells et al., revealed that chronic social defeat stress-induced sleep and circadian rhythm disturbances are attenuated by administering the KOR antagonist JDTic [84]. Zan et al., showed that mice subjected to chronic social defeat stress for 14 days had enhanced dynorphin in the amygdala region. In contrast, KOR-deficient mice and mice microinjected with the KOR antagonist nor-BNI into the amygdala did not show social avoidance induced by chronic social defeat stress, suggesting that the upregulation of dynorphin-KOR in the amygdala is associated with depression-like behavior that occurs following social defeat stress [85]. There is some evidence that stress-induced dynorphin activates the KOR to produce aversion, cocaine preference, and depression-like behaviors. Stress influences KOR expression and its activity in various brain areas such as the VTA, dorsal raphe nucleus, lateral habenula, and NAc. Dynorphin/KOR regulates serotonin activation of 5HT_1B_ receptors within the mNAc and dynamically controls the stress response, affect, and drug reward [86]. On the other hand, Le et al., reported the role of KOR in the bed nucleus of stria terminalis in reinstating alcohol seeking [87]. Ehrich et al., reported that KOR activation of p38α MAPK in VTA dopaminergic neurons was required for conditioned place aversion in mice [88]. Therefore, the endogenous dynorphin-KOR system encodes the dysphoric component of the stress response and controls the risk of depression-like and addiction behaviors [89] as well as motivation and drug- or alcohol-seeking [90]. Dynorphin/KOR activity within the extended amygdala contributes to forced swim stress-enhanced alcohol drinking in mice [91]. Recently, it is reported that aticaprant, a selective and short-acting KOR antagonist, has been under development for the potential treatment of major depressive disorder [92]. Another study showed that aticaprant attenuates alcohol self-administration, reduces alcohol intake escalation, and prevents stress-induced seeking in rodents [11,12]. Moreover, it is suggested that KOR antagonism via aticaprant might prevent alcohol relapse-like drinking in the alcohol deprivation effect model [92,93].

#### 3.1.4. Social Defeat Stress and NOR

The activation of NOR receptors has anxiolytic effects in anxiety-like behavioral tests such as the elevated plus-maze test and the light/dark test. However, N/OFQ knockout (−/−) and NOR−/− mice show higher anxiety levels than wild type [94,95]. On the other hand, NOR inhibitors are known to exhibit antidepressant-like effects [96,97,98]. BTRX-246040, a NOR receptor antagonist with high affinity, potency, and selectivity for NOR receptors, exhibits antidepressant-like effects in forced swimming tests. Bilateral NOR inhibitor administration to the dorsal hippocampal region exhibits antidepressant-like effects in the forced swimming test [99]. Inhibition of NOR signaling may therefore provide an innovative strategy for the treatment of stress-related psychosis [100]. Treatment with the NOR agonist Ro 65-6570 (1 mg/kg) reduced exploration rates in open-field tests for mice subjected to social defeat stress. The NOR agonist also increased immobility and grooming time in the tail suspension test and decreased social interactions [101]. Mice subjected to repeated social defeat stress produce impaired reward learning effects [101]. Repeated social defeat stress causes impaired reward learning in mice. The mechanism involves increased expression of N/OFQ peptide mRNA in the striatum [102]. Acute exposure to social defeat stress in rats enhances NOR receptor expression in the amygdala, but not *prepronociceptin* (*Ppnoc*) mRNA expression [103].

On the other hand, rats exposed to acute restraint stress have increased levels of N/OFQ and NOR mRNA in the amygdala [104]. N/OFQ expression is increased in the hippocampus and dentate gyrus of rats exposed to restraint stress [105]. Clinically, patients diagnosed with postpartum depression, major depressive disorderMDD, and bipolar disorder have been reported to show increased plasma levels of N/OFQ [106,107]. Therefore, the inhibition of MOR or DOR activity may exacerbate emotion-related abnormal behaviors induced by stress. Conversely, the suppression of KOR activity may improve these abnormal behaviors. NOR receptor and N/OFQ expression is increased in response to stressful stimuli, even though the type of stress may differ. Thus, the signaling mechanisms mediated by opioid receptors may show different changes depending on chronic stress (Table 1). 

### 3.2. Influence of Psychosocial Stress on Alteration of Pain Sensitivity and Chronic Pain

Patients with chronic pain often have comorbid emotional dysfunctions, such as depression and anxiety, whereas patients with depression and anxiety have high rates of chronic pain. Rivat et al., revealed that chronic social defeat stress decreases the mechanical pain threshold in rats by increasing inflammatory factors, such as cyclooxygenase-2 and inducible nitric oxide synthase, in the spinal cord [108]. Li et al., show that social defeat stress enhanced plantar incision-induced AMPA receptor GluA1 phosphorylation at the Ser831 site in the spinal cord and greatly prolonged plantar incision-induced pain [109]. Rats exposed to repeated social defeat stress displayed delayed resolution of mechanical hypersensitivity and increased microglial activation and neuronal sensitization within the ipsilateral spinal cord [110]. Suarez-Roca et al., showed that repeated forced swimming-induced stress decreases spinal GABA release and causes hyperalgesia [112]. Accordingly, we hypothesized that psychological stress changes the pain threshold and results in chronic pain, although the detailed mechanism remains unclear.

In contrast, acute stress has an analgesic effect by activating endogenous opioid receptors and other monoamine nervous systems, including the serotonergic and noradrenergic systems. Miczek et al., reported that mice exhibited an analgesic effect against thermal stimulation immediately after exposure to social defeat stress [113]. Coderre et al., claimed that the endogenous opioid b-endorphin is involved in stress-induced analgesia, along with serotonin and noradrenergic neurons [114]. Hohmann et al., demonstrated that two endogenous cannabinoids, 2-arachidonoylglycerol and anandamide, increased in the midbrain after acute stress, indicating that an endocannabinoid mechanism may be involved in stress-induced analgesia [115].

## 4. Involvement of Fatty Acid Receptor Signaling as a Modulator of the Opioid System

We also focused on fatty acid signaling as a possible common factor in the regulation of emotion and pain. We have previously shown that docosahexaenoic acid has antinociceptive effects on various pain stimuli [116], and these effects may be mediated by the incremental release of β-endorphin from pro-opiomelanocortin neurons in the arcuate nucleus of the hypothalamus [117]. GPR40/FFAR1 (G-protein-coupled receptor40/free fatty acid receptor 1) is a GPCR activated by medium- and long-chain fatty acids. This receptor is primarily expressed in humans’ and rodents’ pancreas and central nervous system [118,119,120,121]. Our previous study demonstrated that intracerebroventricular injection of the GPR40/FFAR1 agonist GW9508 or docosahexaenoic acid suppressed formalin-induced pain behavior by increasing the production and release of endorphins in the hypothalamus [122]. GPR40/FFAR1-deficient (GPR40KO) mice showed delayed recovery from post-operative pain [123]. We propose that fatty acid-GPR40/FFAR1 signaling in the brain may be involved in pain control via the regulation of endogenous opioid signaling, particularly endorphin (Figure 2) [124]. We previously reported that GPR40/FFAR1 signaling might regulate the expression of prohormone convertase 2, involved in endorphin production from pro-opiomelanocortin. In contrast, GPR40KO mice exhibited abnormal emotional behaviors, such as a reduction in sucrose preference and anxiety-like behavior (Figure 3) [125]. Our previous study also demonstrated that mice subjected to repeated social defeat stress showed a dramatically longer prolongation of postsurgical pain than non-stressed mice [111]. Social defeat-stressed mice treated with GW1100, a GPR40/FFAR1 antagonist, but not control mice displayed a pronounced prolongation of mechanical allodynia. In addition, mass microscopy imaging analysis showed that phospholipid and fatty acid distribution in some areas, such as the prefrontal cortex and hypothalamus of mice, was significantly reduced after stress exposure [126]. These results suggest that brain fatty acid distribution drastically decreases during exposure to social defeat stress. Reducing fatty acid signaling might disrupt endogenous pain control mechanisms, such as the signaling of endorphin and other opioids, leading to chronic pain. In the future, we would like to determine whether changes in brain fatty acid signaling cause chronic stress-induced pain and explore new drugs targeting fatty acid signaling.

## 5. The Coronavirus Disease Increases Mental Illness and Chronic Pain

The coronavirus disease (COVID-19) pandemic has brought about substantial dynamic changes in family life, society, and health. People have been forced to face many unexpected challenges, which have affected many aspects of life since 2020 [127,128,129]. Children are now more susceptible to psychological and social stress due to various behavioral restrictions in foster care environments. One factor that predisposes individuals to heightened psychological vulnerability during the COVID-19 pandemic is past exposure to adverse childhood experiences [130]. Children are susceptible to psychological and social stress in various situations due to various behavioral restrictions imposed by the COVID-19 pandemic. It may also increase the risk of child abuse. Therefore, a dramatic increase in the number of patients with mental disorders is expected in the future [130,131,132]. Even before the COVID-19 pandemic, it was reported that children who had grown up in an unfavorable nurturing environment were more vulnerable to depression, which became more pronounced during the COVID-19 pandemic [133]. Although there is little evidence, adverse childhood events may have increased due to the COVID-19 pandemic [49,133]. A recent study reported that during the COVID-19 pandemic, there was an increase in the number of children witnessing domestic violence and verbal and emotional abuse [134]. In the near future, the number of children affected by the long-term effects of COVID-19 over the past several years and those who have grown up in unfavorable foster care environments is expected to increase, raising concerns about future health effects. However, there is growing evidence that mental health deterioration is associated with COVID-19 because most people worldwide are under chronic stress. Recent cross-sectional studies in Europe and the US have reported that the COVID-19 pandemic might change the pain threshold and increase the number of patients with chronic pain in the future [135,136,137]. The stressful conditions associated with the COVID-19 pandemic may change opioid function in the central nervous system, resulting in an increased risk of developing psychiatric disorders and chronic pain.

## 6. Conclusions

In this review, we summarized the data on whether ELS or psychosocial stress induces changes in endogenous opioid signaling and leads to the development of psychiatric disorders and chronic pain. There is considerable basic and clinical research evidence that exposure to ELS or psychosocial stress leads to the dysfunction of endogenous opioid signaling and is a risk factor for psychiatric disorders, including depression and anxiety, as well as for the exacerbation and chronicity of pain. Furthermore, our study proposed that fatty acid receptor agonists may suppress emotional and pain behavior via the regulation of endogenous opioid function. Three years have passed since the outbreak of the COVID-19 pandemic and many people are still experiencing stress due to psychosocial factors and changes in the child-rearing environment during childhood. Future research on the effects of ELS and psychological and social stress on opioid signaling will hopefully lead to the development of new preventive methods and treatments for various psychiatric disorders. Identifying children under stress early and providing support and therapeutic interventions are also important.

## Figures and Tables

**Figure 1 ijms-24-11713-f001:**
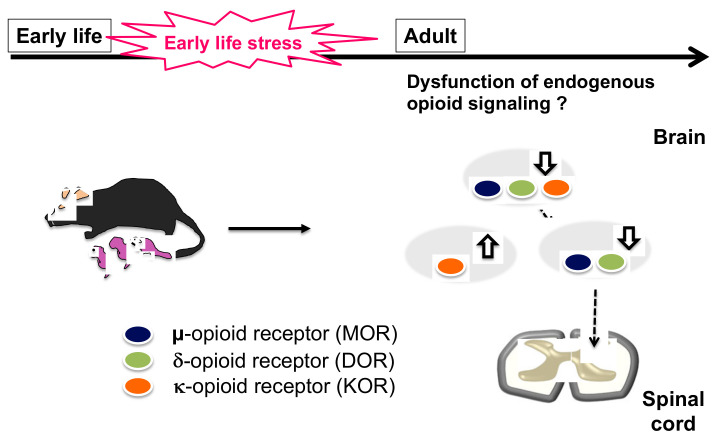
Changes in brain opioid receptors in adulthood after exposure to early life stress. Amy: Amygdala, RVM: Rostral ventrolateral medulla, PAG: Periaqueductal gray. The dotted lines indicate the descending pain control system from PAG to RVM and spinal cord. White arrows indicate increases or decreases in opioid receptor expression.

**Figure 2 ijms-24-11713-f002:**
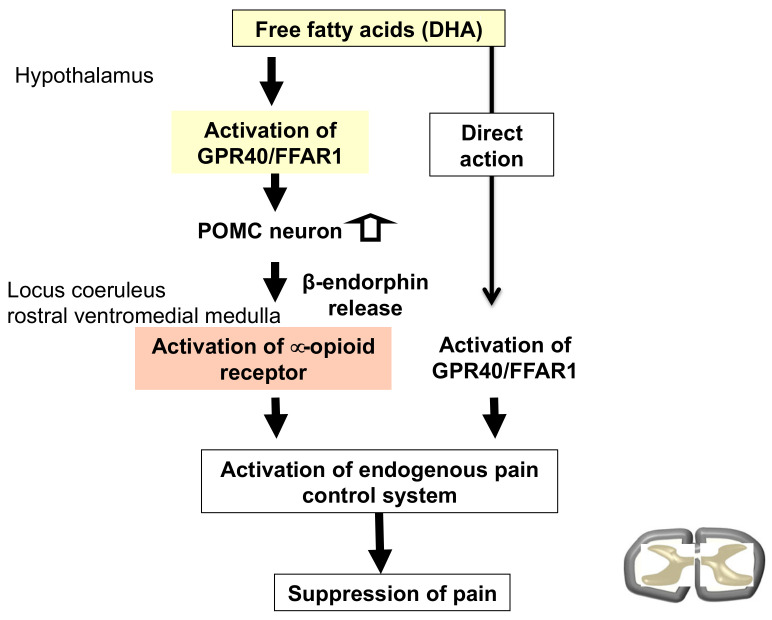
The possible mechanism for DHA-induced antinociception. Downward arrows indicate the flow of antinociceptive mechanisms of DHA. Our hypothesis is that DHA binds GPR40/FFAR1 and directly activates descending pain control mechanisms or indirectly activates these mechanisms via β-endorphin. White arrows represent POMC neuronal activation.

**Figure 3 ijms-24-11713-f003:**
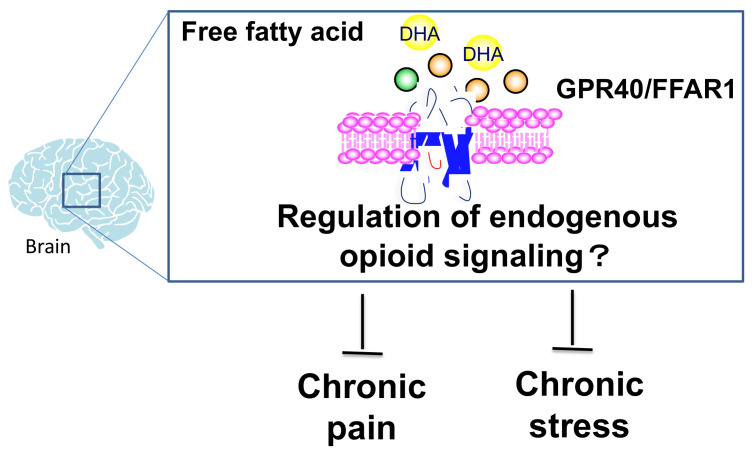
The brain fatty acid-GPR40/FFAR1 signaling may be suppressed to chronic pain and chronic stress via regulation of the endogenous pain control system. Colored circles represent long-chain fatty acids that act on GPR40/FFAR1. “Inverted T-shaped arrows” indicates that GPR40 signaling activation may suppress chronic pain and chronic stress through regulation of endogenous opioid signaling.

**Table 1 ijms-24-11713-t001:** Functional changes of various opioid receptors after exposure of early life stress or social defeat stress in pre-clinical and clinical study.

Behavioral and Functional Changes via Opioid Receptor after Exposure of EARLY life Stress			
Pre-Clinical/Clinical	Model	Changes of Behabioral Phenotype and Each Opioid Receptor Expression in Stressed Mice	References No.
Pre-clinical	ELS model mice induced by maternal separation and social isolation (MSSI)	MSSI mice decreased MOR, DOR, and KOR mRNA expression in the periaqueductal gray matter. In contrast, the expression of KOR mRNA was significantly increased in the amygdala of MSSI mice	[19]
Pre-clinical	Morphine or Naloxone treated mice	Morphine, a MOR agonist, decreased proximity maintenance time in socially housed animals, increases play, whereas naloxonean, an opioid antagonist, reduces play and disrupts pup-retrieval	[27]
Pre-clinical	Naloxone treated mice	Naloxone, an opioid antagonist, decreased play, social behavior and decreased food consumption	[28]
Pre-clinical	MOR knockout mice	MOR knockout mice reduced maternal attachment in mouse pups. Deficits in social behavior	[29]
Pre-clinical	MOR knockout mice	MOR knockout mice decreased social interaction, and exacerbation of anxiety, which are the core and secondary symptoms of autism spectrum disorder	[30,31]
Pre-clinical	Rats with social isolation stress during adolescence	Stressed rats exhibited greater ethanol intake and preference, and induced increases in KOR function contribute to the dysregulation of NAc dopamine	[33]
Pre-clinical	Rats with maternal separation stress	Early postnatal stress can significantly alter the rewarding or aversive value of MOR agonist (morphine) and KOR agonists (spiradoline) when measured using place conditioning	[34]
Pre-clinical	Male and female rats subjected to a neonatal predator odor exposure (POE) paradigm	POE down-regulated neonatal MOR and KOR mRNA levels in neonatal females, but upregulated MOR and DOR mRNA levels in juvenile females, but not male mice	[35]
Pre-clinical	Rats with maternal separation stress	A significant decrease in preproenkephalin mRNA expression was observed in the nucleus accumbens and the caudate-putamen nucleus of deprived rats	[36]
Pre-clinical	The western diet and the environmental stress exposure to the mother during gestation	Mice showed the increased hippocampal *Oprl1* mRNA and *Oprl1* variants	[40]
Clinical	The downregulation of KOR in the anterior insula of Postmortem samples	A history of child abuse was explicitly associated with the downregulation of KOR and decreased DNA methylation in the second intron of Kappa gene in the anterior insula	[18]
Clinical	The child MOR variant OPRM1 A118G	The child MOR variant OPRM1 A118G improved parent-child relationships	[32]
**Early Life Stress and Pain**			
**Pre-Clinical/Clinical**	**Model**	**Changes of Behabioral Phenotype and Each Opioid Receptor** **Expression in Stressed Mice**	**References No.**
Pre-clinical	Mice with maternal separation and social isolation with partial sciatic nerve ligation	MSSI mice with partial sciatic nerve ligation experienced an exacerbation of neuropathic pain compared to non-stressed mice	[19]
Pre-clinical	Mice with maternal separation and social isolation with L5–L6 spinal nerve ligation	Maternal deprivated female, but not male, rats exhibit enhanced nociceptive responding following peripheral nerve injury	[54]
Pre-clinical	Mice with maternal separation and social isolation with partial sciatic nerve ligation	Early Life Adversity as a Risk Factor for Fibromyalgia in Later Life	[55]
Pre-clinical	Mice with maternal separation and social isolation with partial sciatic nerve ligation	Women with a history of ELS have a higher risk of developing irritable bowel syndrome	[56,57]
Clinical	Cross-sectional analysis of the 2016–2017 National Survey of Children’s Health	Children with exposure to 1 or more ACEs had higher rates of chronic pain as compared to those with no reported ACEs. Children and adolescents with ACEs had increased risk for chronic pain	[2]
Clinical	Data from the 1958 British Birth Cohort Study	Children who had resided in institutional care experienced an increase in the risk of chronic widespread pain at adulthood as did those who experienced maternal death and familial financial hardship	[5]
Clinical	Self-report (Childhood Trauma Questionnaire-short form	In women with chronic pain, self-reported childhood maltreatment was associated with higher diurnal cortisol levels	[49]
Clinical	Eighty-eight females in A Cross-Sectional Study	Fibromyalgia patients having suffered traumatic events throughout their lifespan, especially in childhood and early adolescence. All patients showed clinically relevant levels of anxiety, depression, insomnia, suicidal thoughts, and pain, as well as somatic comorbidities and poor quality of life	[50]
Clinical	A sample of mothers with chronic pain and their 8- to 12-year-old children in A Cross-Sectional Study	Higher maternal adverse childhood experiences (ACEs) scores corresponded with depressive symptoms and greater anxiety, greater sleep disturbances and fatigue, and greater pain intensity in mothers	[51]
Clinical	Meta analysis review	Children who experienced childhood abuse and neglect had worse pain symptoms than those who did not experience early life stress (ELS)	[52]
**Social Defeat Stress and Opioid**			
**Pre-Clinical/Clinical**	**Model**	**Changes of Behabioral Phenotype and Each Opioid Receptor** **Expression in Stressed Mice**	**References No.**
Pre-clinical	Sprague-Dawley rats	Enkephaline mRNA expression was lower in the basolateral nucleus of the amygdala in vulnerable. Dynorphin mRNA is increased only within the dorsal and medial shell of the NAc of vulnerable rats	[20]
Pre-clinical	Rats exposed to restraint stress	N/OFQ and NOR mRNA expression is increased in the hippocampus	[37]
Pre-clinical	Mouse model of *Oprm1* (A118G) polymorphism	Mouse model of *Oprm1* (A118G) polymorphism increased sociability and dominance and confers resilience to social defeat	[61]
Pre-clinical	Lentivirus-mediated MOR knockdown in the VTA	Social defeat stress increased MOR mRNA expression and leads to an increased capacity for MOR activation in the ventral tegmental area (VTA). Social stress exposure induced social avoidance were prevented by VTA MOR knockdown	[62,63,64]
Pre-clinical	MOR deficient mice	MOR deficient mice did not exhibit socially aversive behavior and reduced activity of the hypothalamic-pituitary-adrenocortical axis associated with the stress response and improvement in symptoms such as stress-induced depression-like behavior	[72]
Pre-clinical	Mice with DOR agonist or antagonist	DOR agonists showed anxiolytic and antidepressant effects, whereas DOR antagonists or DOR deficient mice exhibited enhanced anxiety-like behaviors	[73,74,75,76]
Pre-clinical	Adult C57BL/6J mice	SDS-induced susceptible mice showed reduced enkephalin levels in the NAc. DOR agonist, SNC80, improved sds-induced reduction of social behavior	[79]
Pre-clinical	Adult C57BL/6J mice	Repeated administrations to cVSDS mice with a selective DOP agonist, KNT-127 improved their decreased social interaction behaviors and increased serum corticosterone levels.	[80]
Pre-clinical	Adult C57BL/6J mice	KOR antagonist nor-binaltorphimine blocked the stress-induced analgesia, and significantly reduced the stress-induced immobility on the second and third days of SDS exposure. In contrast, prodynorphin gene-disrupted mice improved these effects	[83]
Pre-clinical	Adult C57BL/6J mice	KOR antagonist JDTic reduced stress effects on both sleep and circadian rhythms	[84]
Pre-clinical	Adult C57BL/6J mice	KOR antagonist norBNI or KOR knockout mice could prevent the development of social avoidance induced by chronic social defeat stress	[85]
Pre-clinical	N/OFQ knockout (−/−) and NOR−/− mice	N/OFQ knockout (−/−) and NOR−/− mice showed higher anxiety behavior	[94,95]
Pre-clinical	Adult C57BL/6J mice	NOR inhibitors are known to exhibit antidepressant-like effects	[96,97,98,99]
Pre-clinical	Acute exposure to social defeat stress in rats	Stressed mice enhanced NOR receptor expression in the amygdala	[103]
Clinical	Patients diagnosed with postpartum depression, major depressive disorder and bipolar disorder	Patientss showed increased plasma levels of N/OFQ	[106,107]
**Social Defeat Stress and Pain**			
**Pre-Clinical/Clinical**	**Model**	**Changes of Behabioral Phenotype and Each Opioid Receptor** **Expression in Stressed Mice**	**References No.**
Pre-clinical	Social defeat stress model mice with incision pain	Social defeat stress decreases the mechanical pain threshold. Increasing inflammatory factors, such as COX-2 and iNOS, in the spinal cord.	[108]
Pre-clinical	Social defeat stress model mice	Social defeat stress enhanced plantar incision-induced AMPA receptor GluA1 phosphorylation at the Ser831 site in the spinal cord.	[109]
Pre-clinical	Social defeat stress model mice	Social defeat stress displayed delayed resolution of mechanical hypersensitivity and increased micro-glial activation and neuronal sensitization within the ipsilateral spinal cord.	[110]
Pre-clinical	Social defeat stress model mice	Mice subjected to repeated social defeat stress showed a dramatically longer prolongation of postsurgical pain through decreasing brain fatty acid signaling.	[111]

## Data Availability

Data sharing is not applicable to this article.

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
