# Peer review of "Stress-Induced Changes in the Endogenous Opioid System Cause Dysfunction of Pain and Emotion Regulation"

_ijms, 2023, doi:10.3390/ijms241411713_

Round 1

Reviewer 1 Report

The current review aims to discuss the literature regarding the involvement of the opioid system in stress-mediated pain chronicity and neuropsychiatric disorders. The authors discussed changes in the levels of opioid peptides/receptors in response to stress or in animal models of stress as well as in mice deficient of the receptors or peptides and agonists and antagonists of each receptor system. Overall, the review captures a nice wealth of data and is well organized. However, I have a few comments that need to be addressed.

Major:

1. Given no evidence of alterations in opioids and opioid receptors in response to COVID-19 exists, I feel this section may be a bit out of place. Still, I leave this to the authors and the editor to decide whether it should remain in the revised manuscript.

2. The authors discuss chronicity of pain, yet only provide evidence showing alterations in neuropathic pain models. Does ELS affect other form of chronic pain? Please address this.

3. If MOP-deficient mice show reduced depression-like behaviors, MOP makes them prone or resilient? (line 251).

Minor:

1. Please use either MOR or MOP throughout the manuscript.

2. Please correct line 268; there may be unnecessary information following basal ganglia.

3. There is a study where only beta-endorphin was shown to be involved in stress-induced analgesia (Parikh et al., EJP 2011) and can be relevant to the current review to be included. 

Author Response

Reviewer 1

We would like to thank the reviewer’s comment and suggestion.

  1. Given no evidence of alterations in opioids and opioid receptors in response to COVID-19 exists, I feel this section may be a bit out of place. Still, I leave this to the authors and the editor to decide whether it should remain in the revised manuscript.

(Answer) Thank you for your comments. As you pointed out, as far as we know, there is no evidence of altered opioid receptor function or its expression in the presence of COVID-19 at this time. However, there is growing evidence regarding increased mortality caused by synthetic opioid and fentanyl overdose in the United States since the COVID-19 pandemic [Brown et al., 2022], suggesting that the coronavirus may have caused functional alterations of opioid receptors. Given the current lack of convergence of COVID-19, future studies will be required to clear the interaction between COVID-19 and functional alterations of opioid receptors. Therefore, we propose our hypothesis and wish to leave this section in our manuscript.

Brown et al., Has the United States Reached a Plateau in Overdoses Caused by Synthetic Opioids After the Onset of the COVID-19 Pandemic? Examination of Centers for Disease Control and Prevention Data to November 2021. Front Psychiatry. 2022 Jul 7;13:947603.

  1. The authors discuss the chronicity of pain, yet only provide evidence showing alterations in neuropathic pain models. Does ELS affect other forms of chronic pain? Please address this.

(Answer) Thank you for your meaningful comment. As you pointed out, ELS is known to exacerbate not only neuropathic pain but also fibromyalgia [55] and abdominal pain which associate with irritable bowel syn-drome (IBS) [56, 57]. However, there are few reports on the involvement of altered function of opioid receptors and changes in their receptor ex-pression in the mechanism for ELS-induced fibromyalgia and IBS-induced abdominal pain.

  1. Low, L. A.; Schweinhardt, P., Early life adversity as a risk factor for fibromyalgia in later life. Pain Res Treat 2012, 2012, 140832.
  2. Louwies, T.; Mohammadi, E.; Greenwood-Van Meerveld, B., Epigenetic mechanisms underlying stress-induced visceral pain: Resilience versus vulnerability in a two-hit model of early life stress and chronic adult stress. Neurogastroenterol Motil 2023, 35, (5), e14558.
  3. Louwies, T.; Greenwood-Van Meerveld, B., Sex differences in the epigenetic regulation of chronic visceral pain following unpredictable early life stress. Neurogastroenterol Motil 2020, 32, (3), e13751.

  1. If MOP-deficient mice show reduced depression-like behaviors, MOP makes them prone or resilient? (line 251).

(Answer) Thank you for your comment, Ide et al., suggest that  MOP play an important facilitatory role in emotional responses to stress, including anxiety- and depression-like behavior and corticosterone levels (19596019). So, it is thought that modulation of MOP might show resilience against stress-induced anxiety or depression.

Ide S, Fujiwara S, Fujiwara M, Sora I, Ikeda K, Minami M, Uhl GR, Ishihara K. Antidepressant-like effect of venlafaxine is abolished in μ-opioid receptor-knockout mice. J Pharmacol Sci. 2010;114(1):107-10.

Minor

  1. Please use either MOR or MOP throughout the manuscript.

(Answer) We used MOR throughout the manuscript.

  1. Please correct line 268; there may be unnecessary information following basal ganglia.

(Answer) We revised the word “basal ganglia of the amygdala” to “basolateral amygdala” as follows. Enkephalin mRNA expression is decreased in the basolateral amygdala (BLA) and ventral hippocampal CA1 region of stress-vulnerable mice after social defeat stress exposure. Thus, activating the enkephalin/DOP signaling pathway may have a preventive effect on stress vulnerability caused by psychological stress.

  1. There is a study where only beta-endorphin was shown to be involved in stress-induced analgesia (Parikh et al., EJP 2011) and can be relevant to the current review to be included. 

(Answer)  We included the paper you mentioned in the references list.

Parikh et al., Stress-induced analgesia and endogenous opioid peptides: the importance of stress duration. Eur J Pharmacol. 2011;650(2-3):563-7. 

Reviewer 2 Report

The studies described in the review article "Stress-Induced Changes in the Endogenous Opioid System cause Dysfunction of Pain and Emotion Regulation" by Nakamoto and Tokuyama summarize studies on the effects of endogenous opioid signaling on early-life stress, psychosocial stress-induced psychiatric disorders, and chronic pain.

The following are some suggestions to improve the article:

1.       Line 69-72: Please cite a reference

2.       Line 147-149: Please rephrase for better understanding.

3.       Figure 1: Please expand the acronyms (Amy, PAG, RVM) used in figure 1 in the captions. What do the dotted lines represent? Also, drawing a boundary of the brain showing those 3 brain areas will be helpful to readers.

4.       The authors should follow a consistent pattern of writing a gene/protein nomenclature.

- For humans, the gene symbol should be capitalized and italicized (OPRM1). Protein symbols are the same as gene symbols but should not be italicized.

-For animals (mouse, rat, and chicken), the gene symbols are italicized, first letter upper case and all the rest lower case (Oprm1). Protein designations are the same as gene symbols, but not italicized and all are upper case.

5.       For the receptors also, peers preferentially use KOR, MOR and DOR in place of KOP, MOP and DOP.

6.       3.1.1 Social Defeat Stress and MOP section on Page 6: Which cell population in VTA expresses MOR? This information is important while explaining the effects of stress on MOR as this can provide important information about the circuitry involved.

7.       The authors might consider discussing “Psychoneuroendocrinology 2008 May;33(4):478-86” for acute effects of MORs on the HPA axis.

8.       Line 225-265: Please cite references (original research articles) corresponding to each behavioral effect mentioned in the statement.

9.       Line 268: What are the "basal ganglia of the amygdala"? “Enkephalin mRNA expression is decreased in the basal ganglia of the amygdala (BLA) and ventral hippocampal CA1 region of stress-vulnerable mice after social defeat stress exposure.” BLA generally depicts the basolateral amygdala.

10.   It will be best to explain the basal ganglia circuitry and its role briefly before mentioning the effect of stress on opioid receptors in the areas of basal ganglia.

11.   Stress and KOR section: The authors mentioned only one study related to KOR and the amygdala. How about KORs in VTA, Dorsal Raphe (seminal studies from Charles Chavkin group), lateral Habenula, and Nucleus accumbens? Role of KORs in stress-induced binge drinking, and cocaine intake?

12.   Figures 2 and 3: Figure legends should have information related to the figures.

13. KOR antagonists were in clinical trials for depression. These findings should also be mentioned in the manuscripts. 

14. A table mentioning the preclinical/clinical studies along with references related to each receptor will be helpful.

The English language is fine.

Author Response

IJMS revise

Reviewer 2

We would like to thank the reviewer’s comment and suggestion.

The following are some suggestions to improve the article:

  1. Line 69-72: Please cite a reference

(Answer) We inserted references in the manuscript. (Line 73)

  1. Line 147-149: Please rephrase for better understanding.

(Answer) Thank you for your advice. As you pointed out, we revised this sentence. (Line 150-152)

  1. Figure 1: Please expand the acronyms (Amy, PAG, RVM) used in figure 1 in the captions. What do the dotted lines represent? Also, drawing a boundary of the brain showing those 3 brain areas will be helpful to readers.

(Answer) In the caption, we inserted the acronyms (Amy, PAG, RVM) in Fig. 1. We showed the boundary of the three brains area in Fig. 1. The dotted lines indicate the descending pain control system from PAG to RVM. (Line 178-179)

  1. The authors should follow a consistent pattern of writing a gene/protein nomenclature. - For humans, the gene symbol should be capitalized and italicized (OPRM1). Protein symbols are the same as gene symbols but should not be italicized. -For animals (mouse, rat, and chicken), the gene symbols are italicized, first letter upper case and all the rest lower case (Oprm1). Protein designations are the same as gene symbols, but not italicized and all are upper case.

(Answer) Thank you for your meaningful suggestion. We revised the gene symbols as you pointed out throughout the manuscript.

  1. For the receptors also, peers preferentially use KOR, MOR and DOR in place of KOP, MOP and DOP.

(Answer) We used KOR, DOR, and MOR in the manuscript.

  1. 3.1.1 Social Defeat Stress and MOP section on Page 6: Which cell population in VTA expresses MOR? This information is important while explaining the effects of stress on MOR as this can provide important information about the circuitry involved.

(Answer) Nikulina et al. reported that social defeat stress upregulated MOR expression in the VTA area. Generally, MOR activation inhibits GABA neurons to distinct dopamine neuron. In this study, repeated social defeat stress exposure induced social avoidance in wild-type mice, but these effects were prevented by the pretreatment of VTA MOR knockdown. In Jonston’s paper, unfortunately, it is not clear which cells express MOR in the VTA region. Kudo et al. have been reported that 74.1% of MOR mRNA-expressing neurons in the VTA were GABAergic, with the rest being glutamatergic neurons expressing type-2 vesicular glutamate transporter mRNA [65]. VTA MORs are presynaptically expressed by GABA neurons [66, 67] and when activated, reduce GABAergic inhibition of VTA DA neurons [68] in the VTA. Social stressors may function to increase endogenous MOR activity on GABA neurons, thus reducing the GABAergic inhibition of local DA neurons and facilitating behavioral sensitization to psychostimulant drugs. Further study will be needed to clarify which cells express MOR in the VTA using genetic MOR-Cre mice.

  1. Kudo, T.; Konno, K.; Uchigashima, M.; Yanagawa, Y.; Sora, I.; Minami, M.; Watanabe, M., GABAergic neurons in the ventral tegmental area receive dual GABA/enkephalin-mediated inhibitory inputs from the bed nucleus of the stria terminalis. Eur J Neurosci 2014, 39, (11), 1796-809.
  2. Garzon, M.; Pickel, V. M., Ultrastructural localization of enkephalin and mu-opioid receptors in the rat ventral tegmental area. Neuroscience 2002, 114, (2), 461-74.
  3. Sesack, S. R.; Pickel, V. M., Ultrastructural relationships between terminals immunoreactive for enkephalin, GABA, or both transmitters in the rat ventral tegmental area. Brain Res 1995, 672, (1-2), 261-75.
  4. Dacher, M.; Nugent, F. S., Morphine-induced modulation of LTD at GABAergic synapses in the ventral tegmental area. Neuropharmacology 2011, 61, (7), 1166-71.
  5. The authors might consider discussing “Psychoneuroendocrinology 2008 May;33(4):478-86” for acute effects of MORs on the HPA axis.

(Answer) Thank you for your suggestion. Previously, human and primate studies reported that MOR agonists suppress HPA axis [69, 70], whereas rodent studies showed that MOR ago-nists increased plasma adrenocorticotropic hormone (ACTH) and corti-costerone levels [71]. There seem to be numerous contradictions in the lit-erature regarding the effects of opioids on the HPA axis. We inserted these points into the manuscript.

  1. Pascoe, J. E.; Williams, K. L.; Mukhopadhyay, P.; Rice, K. C.; Woods, J. H.; Ko, M. C., Effects of mu, kappa, and delta opioid receptor agonists on the function of hypothalamic-pituitary-adrenal axis in monkeys. Psychoneuroendocrinology 2008, 33, (4), 478-86.
  2. Hoehe, M.; Duka, T.; Doenicke, A., Human studies on the mu opiate receptor agonist fentanyl: neuroendocrine and behavioral responses. Psychoneuroendocrinology 1988, 13, (5), 397-408.
  3. Iyengar, S.; Kim, H. S.; Wood, P. L., Mu-, delta-, kappa- and epsilon-opioid receptor modulation of the hypothalamic-pituitary-adrenocortical (HPA) axis: subchronic tolerance studies of endogenous opioid peptides. Brain Res 1987, 435, (1-2), 220-6.
  4. Line 225-265: Please cite references (original research articles) corresponding to each behavioral effect mentioned in the statement.

(Answer) We cite original research articles in the reference list.

  1. Line 268: What are the "basal ganglia of the amygdala"? “Enkephalin mRNA expression is decreased in the basal ganglia of the amygdala (BLA) and ventral hippocampal CA1 region of stress-vulnerable mice after social defeat stress exposure.” BLA generally depicts the basolateral amygdala.

(Answer) We are sorry about that. We mistook the word. According to your suggestion, we changed the word “the basal ganglia of the amygdala (BLA)” to “the basolateral amygdala (BLA)”.

  1. It will be best to explain the basal ganglia circuitry and its role briefly before mentioning the effect of stress on opioid receptors in the areas of basal ganglia.

(Answer) The word of  “basal ganglia” was deleted in our manuscript.

  1. Stress and KOR section: The authors mentioned only one study related to KOR and the amygdala. How about KORs in VTA, Dorsal Raphe (seminal studies from Charles Chavkin group), lateral Habenula, and Nucleus accumbens? Role of KORs in stress-induced binge drinking, and cocaine intake?

(Answer) We appreciated your comments.

There is some evidence that stress-induced dynorphin activates the KOR to produce aversion, cocaine preference and depression-like behaviors. Stress influ-ences KOR expression and its activity in the various brain areas such as the VTA, dorsal raphe nucleus (DRN), lateral habenula and NAc. Dyn/KOR regulates sero-tonin activation of 5HT1B receptors within the mNAc and dynamically controls stress response, affect, and drug reward [86]. On the other hand, Le et al. reported the role of KOR in the Bed Nucleus of Stria Terminalis (BNST) in reinstating al-cohol seeking [87]. Ehrich et al. reported that we report that KOR activation of p38α MAPK in ventral tegmental (VTA) dopaminergic neurons was required for conditioned place aversion in mice [88]. Therefore, the endogenous dynorphin-KOR system encodes the dysphoric component of the stress response and controls the risk of depression-like and addiction behaviors [89] as well as motivation and drug- or alcohol-seeking [90]. Dynorphin/KOR activity within the extended amygdala contributes to forced swim stress-enhanced alcohol drinking in mice [91].

  1. Fontaine, H. M.; Silva, P. R.; Neiswanger, C.; Tran, R.; Abraham, A. D.; Land, B. B.; Neumaier, J. F.; Chavkin, C., Stress decreases serotonin tone in the nucleus accumbens in male mice to promote aversion and potentiate cocaine preference via decreased stimulation of 5-HT(1B) receptors. Neuropsychopharmacology 2022, 47, (4), 891-901.
  2. Le, A. D.; Funk, D.; Coen, K.; Tamadon, S.; Shaham, Y., Role of kappa-Opioid Receptors in the Bed Nucleus of Stria Terminalis in Reinstatement of Alcohol Seeking. Neuropsychopharmacology 2018, 43, (4), 838-850.
  3. Ehrich, J. M.; Messinger, D. I.; Knakal, C. R.; Kuhar, J. R.; Schattauer, S. S.; Bruchas, M. R.; Zweifel, L. S.; Kieffer, B. L.; Phillips, P. E.; Chavkin, C., Kappa Opioid Receptor-Induced Aversion Requires p38 MAPK Activation in VTA Dopamine Neurons. J Neurosci 2015, 35, (37), 12917-31.
  4. Kendler, K. S.; Karkowski, L. M.; Prescott, C. A., Causal relationship between stressful life events and the onset of major depression. Am J Psychiatry 1999, 156, (6), 837-41.
  5. Bruchas, M. R.; Land, B. B.; Chavkin, C., The dynorphin/kappa opioid system as a modulator of stress-induced and pro-addictive behaviors. Brain Res 2010, 1314, 44-55.
  6. Haun, H. L.; Lebonville, C. L.; Solomon, M. G.; Griffin, W. C.; Lopez, M. F.; Becker, H. C., Dynorphin/Kappa Opioid Receptor Activity Within the Extended Amygdala Contributes to Stress-Enhanced Alcohol Drinking in Mice. Biol Psychiatry 2022, 91, (12), 1019-1028.

  1. Figures 2 and 3: Figure legends should have information related to the figures.

(Answer) Thank you for your comments. We revised Figure legend 2 and 3.

  1. KOR antagonists were in clinical trials for depression. These findings should also be mentioned in the manuscripts. 

(Answer) Thank you for your meaningful suggestion. Recently, it is reported that aticaprant, a selective and short-acting KOR antagonist, has been under development for the potential treatment of major depressive disorder [92]. Another study showed that aticaprant attenuates alcohol self-administration, reduces alcohol intake escalation, and prevents stress-induced seeking in rodents [11,12]. Also, it is provided that KOR antagonism aticaprant might prevent alcohol relapse-like drinking in the alcohol deprivation effect model [92, 93].

  1. Krystal, A. D.; Pizzagalli, D. A.; Smoski, M.; Mathew, S. J.; Nurnberger, J., Jr.; Lisanby, S. H.; Iosifescu, D.; Murrough, J. W.; Yang, H.; Weiner, R. D.; Calabrese, J. R.; Sanacora, G.; Hermes, G.; Keefe, R. S. E.; Song, A.; Goodman, W.; Szabo, S. T.; Whitton, A. E.; Gao, K.; Potter, W. Z., A randomized proof-of-mechanism trial applying the 'fast-fail' approach to evaluating kappa-opioid antagonism as a treatment for anhedonia. Nat Med 2020, 26, (5), 760-768.
  2. Rorick-Kehn, L. M.; Witkin, J. M.; Statnick, M. A.; Eberle, E. L.; McKinzie, J. H.; Kahl, S. D.; Forster, B. M.; Wong, C. J.; Li, X.; Crile, R. S.; Shaw, D. B.; Sahr, A. E.; Adams, B. L.; Quimby, S. J.; Diaz, N.; Jimenez, A.; Pedregal, C.; Mitch, C. H.; Knopp, K. L.; Anderson, W. H.; Cramer, J. W.; McKinzie, D. L., LY2456302 is a novel, potent, orally-bioavailable small molecule kappa-selective antagonist with activity in animal models predictive of efficacy in mood and addictive disorders. Neuropharmacology 2014, 77, 131-44.
  3. A table mentioning the preclinical/clinical studies along with references related to each receptor will be helpful.

(Answer)We appreciate your meaningful comment. We created Table 1 which showed the preclinical/clinical studies along with references related to each  receptor.

Round 2

Reviewer 2 Report

1. Line 103: It should be OPRM1 A118G.

2. Line 110- Please rephrase- "socially isolated rats".

3. Line 136- learning "and " memory.

4. Line 252: using MOR conditional knock-out mice. 

5. Line 260- Looks like the authors forgot to delete " We inserted these points into the manuscript."

6. Line 297- administrations "of"

7. Line 355- defeat stress "show"

8. I do not see Table-1 in the manuscript.

I have pointed out a few issues in the manuscript.

Author Response

Again, we would like to thank the reviewer’s comment and suggestion.

1. Line 103: It should be OPRM1 A118G.

(Answer) We changed “OPRM1” to “OPRM1

2. Line 110- Please rephrase- "socially isolated rats".

(Answer) We changed “Social isolation rats ” to  "socially isolated rats".

3. Line 136- learning "and " memory.

(Answer) We revised it.

4. Line 252: using MOR conditional knock-out mice. 

(Answer) We revised it.

5. Line 260- Looks like the authors forgot to delete " We inserted these points into the manuscript."

(Answer) We deleted this word.

6. Line 297- administrations "of"

(Answer) We deleted it.

7. Line 355- defeat stress "show"

(Answer) We revised it.

8. I do not see Table-1 in the manuscript.

(Answer) We showed the position of Table 1 in the manuscript